# Comparison study of image quality at various radiation doses for CT venography using advanced modeled iterative reconstruction

Jung Han Hwang[1☯‡], Jin Mo Kang[2☯‡], So Hyun Park[1]*, Suyoung Park[1], Jeong Ho Kim[1], Sang tae Choi[2]

1 Department of Radiology, Gil Medical Center, Gachon University College of Medicine, Incheon, Republic of Korea, 2 Department of Surgery, Gil Medical Center, Gachon University College of Medicine, Incheon, Republic of Korea

☯ These authors contributed equally to this work.
‡ These authors share first authorship on this work.
* nnoleeter@naver.com

## Abstract

### Objective

We compared the image quality according to the radiation dose on computed tomography (CT) venography at 80 kVp using advanced modeled iterative reconstruction for deep vein thrombus and other specific clinical conditions considering standard-, low-, and ultralow-dose CT.

### Methods

In this retrospective study, 105 consecutive CT venography examinations were included using a third-generation dual-source scanner in the dual-source mode in tubes A (reference mAs, 210 mAs at 70%) and B (reference mAs, 90 mAs at 30%) at a fixed 80 kVp. Two radiologists independently reviewed each observation of standard- (100% radiation dose), low- (70%), and ultralow-dose (30%) CT. The objective quality of large veins and subjective image quality regarding lower-extremity veins and deep vein thrombus were compared between images according to the dose. In addition, the CT dose index volumes were displayed from the images.

### Results

From the patients, 24 presented deep vein thrombus in 69 venous segments of CT examinations. Standard-dose CT provided the lowest image noise at the inferior vena cava and femoral vein compared with low- and ultralow-dose CT ($p < 0.001$). There were no differences regarding subjective image quality between the images of popliteal and calf veins at the three doses (e.g., 3.8 ± 0.7, right popliteal vein, $p = 0.977$). The image quality of the 69 deep vein thrombus segments showed equally slightly higher scores in standard- and low-dose CT (4.0 ± 0.2) than in ultralow-dose CT (3.9 ± 0.4). The CT dose index volumes were 4.4 ± 0.6, 3.1 ± 0.4, and 1.3 ± 0.2 mGy for standard-, low-, and ultralow-dose CT, respectively.

**Data Availability Statement:** The data for this study contain potentially identifying information. The data are available from the Data Access Gil

hospital contact via Hyun Joo Lee (lhj@gilhospital.com).

**Funding:** This research was supported by the Basic Science Research Program through the National Research Foundation of Korea and funded by the Ministry of Science ICT and Future Planning (2018R1C1B5044024).

**Competing interests:** The authors have declared that no competing interests exist.

## Conclusions

Low- and ultralow-dose CT venography at 80 kVp using an advanced model based iterative reconstruction algorithm allows to evaluate deep vein thrombus and perform follow-up examinations while showing an acceptable image quality and reducing the radiation dose.

## Introduction

Venous thromboembolism is the third main cause of cardiovascular disease [1], and its incidence has sharply increased over the last two decades [2]. It occurs in two forms, deep vein thrombosis (DVT) and pulmonary embolism. DVT is often related with recurrent venous thromboembolism and pulmonary embolism according to the disease process [3]. Disease recurrence occurs in 20–36% of the DVT patients as the disease progresses [4, 5]. Chronic venous change, venous bleeding, and death are the major consequences that may occur during the clinical course of DVT. Along with ultrasonography, computed tomography (CT) venography of the lower extremity is common for DVT diagnosis and follow-up.

There are many studies in the literature investigating radiation dose reduction with low tube voltages and advanced model based reconstruction [6–10]. We are contributing to this space by specifically looking at the image quality of DVT segments, chronic venous change, stent placement, and metal artifacts affecting the vein segments on low- and standard-dose CT venography. Nevertheless, such conditions are often encountered in clinical practice when radiologists review CT venograms.

Iterative reconstruction has been developed using statistical algorithms, and model-based iterative reconstruction algorithms have been recently introduced [11]. In addition, advanced modeled iterative reconstruction (ADMIRE; Siemens Healthineers, Forchheim, Germany) is a model-based algorithm that decreases raw data noise and enables radiation dose reduction with maintaining the image quality of CT scans. Dual-source CT scanners can blend or divide raw data acquired from each tube, allowing the generation of images at different radiation doses in a single CT examination [12, 13]. In this study, we compared the image quality according to radiation dose on CT venography at 80 kVp using ADMIRE regarding specific clinical conditions and considering standard-, low-, and ultralow-dose CT that using ADMIRE promotes dose reduction while maintaining the image quality.

## Materials and methods

### Study design

This retrospective study was approved by the Gil Medical Center institutional review board. The requirement for informed consent was waived given the retrospective nature of this study. The CT scans were performed using standard-dose radiation without additional dose exposure.

### Patients

One hundred ten CT venography examinations were performed in a tertiary care center for either DVT diagnosis or follow-up between May 2019 and September 2020. The CT protocol of 5 examinations was different from that of the others and excluded from this study. Thus, 105 examinations from 100 patients (48 men, 52 women; mean age, 63.5 years; 18–94 years) were considered (**Fig 1A**). The clinical characteristics of the patients are listed in **Table 1**.

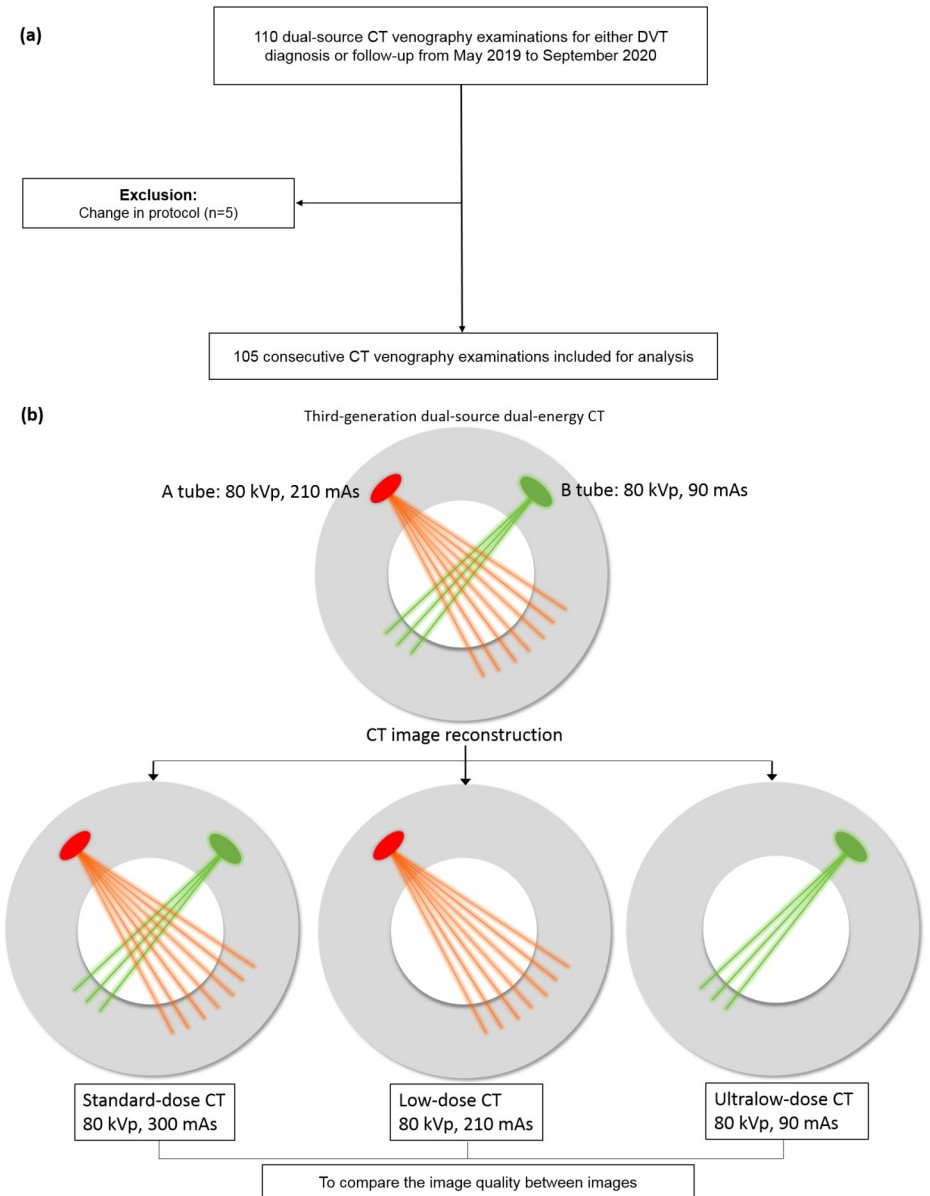

**Fig 1. Flowchart of patient inclusion.** (a) Inclusion process. (b) study design.

## Protocol

The patients underwent CT venography examinations from the T12 vertebra to the feet. To obtain the contrast-enhanced images, 1.5 mL/kg according to the body weight (maximum, 150 mL) of contrast media (Iohexol 350 mgI/mL—Bonorex 350; Central Medical Services, Seoul, Republic of Korea) at a flow rate of 3 mL/s was injected in each patient followed by 30 mL of 0.9% saline solution at the same flow rate. The CT scans were performed with a 192-slice CT scanner (SOMATOM Force; Siemens Healthineers, Erlangen, Germany) in the dual-source mode in tubes A (reference mAs, 210 mAs at 70%) and B (reference mAs, 90 mAs at 30%) using tube current modulation (CARE Dose 4D, Siemens Healthineers) at a fixed 80 kVp tube voltage in **Fig 1B**. The pitch was 0.6, and the rotation time was 0.5 s. The images were obtained

**Table 1. Patient characteristics.**

| Parameter | Value |
|---|---|
| **Patients (Male)** | 100 (48) |
| **Age (years)** | 63.5 ± 16.2 |
| **Height (cm)** | 162.1 ± 10.6 |
| **Weight (kg)** | 65.5 ± 13.6 |
| **Body mass index** | 24.8 ± 4.0 |
| **<18.5 (underweight)** | 10 |
| **18.5–24.9 (normal)** | 41 |
| **25–29.9 (overweight)** | 40 |
| **30–34.9 (moderately obese)** | 8 |
| **35–39.9 (severely obese)** | 1 |

Note. Data are means ± standard deviations.

using standard- (A and B tube data), low- (tube A data), and ultralow-dose (tube B data) CT, obtaining three image sets. To produce the specific split of the radiation dose (mAs) between each tube detector, the CT scanner needs a dual energy research license. The images were reconstructed with axial slice thickness of 5 mm using ADMIRE at strength level 3. From the reports in [3] and our preliminary examinations between March and April 2019, we designed the CT dose index volume ($CTDI_{vol}$) at the ultralow dose to be approximately 1.5 mGy.

## Data analysis

All image analyses were performed using a picture archiving and communication system (PACS). The CT scans were independently reviewed by two radiologists with 10 and 13 years of experience. Diverging interpretations were reevaluated by the radiologists to reach a consensus. The 315 images (105 examinations × 3 image sets) were analyzed for the three dose levels with a washout period (6 weeks).

## Subjective image quality analysis

A subjective image quality analysis was performed by the two radiologists, who were blinded to the radiation dose and patient's information. The overall CT image quality and the segment image quality of the inferior vena cava (IVC), bilateral common iliac veins (CIVs), bilateral femoral veins (FVs), bilateral popliteal veins, and bilateral calf veins were scored using the following four-point scale: 1) poor, unacceptable subjective image noise with artifacts impeding diagnosis; 2) adequate, average image noise and acceptable information for diagnosis; 3) good, low image noise and necessary information for adequate diagnosis; and 4) excellent, very low image noise and optimal information for diagnosis [12]. A score of 1 was regarded unacceptable for diagnosis. Analogously, the venous contrast was graded using the following four-point scale: 1) poor, enhancement below adjacent muscular enhancement; 2) adequate, enhancement similar to surrounding muscle enhancement; 3) good, inhomogeneous enhancement, less intense than the corresponding artery but more than surrounding muscle; and 4) excellent, homogenous enhancement similar to the corresponding arterial enhancement.

The following conditions were analyzed regarding the evaluations and image quality: 1) acute DVT of lower-extremity vein segment, 2) May–Thurner syndrome, 3) chronic venous change, 4) in-stent restenosis in patients with uncovered or covered stent, 5) artifacts due to prosthesis, and 6) incidental findings (e.g., varicose vein). Acute DVT was diagnosed by the presence of complete or partial low-attenuation intraluminal filling defects on CT venograms

for at least two consecutive axial images [14]. Chronic venous change (i.e., chronic-stage DVT) was diagnosed by the presence of decreased vessel caliber, fibrotic bands, recanalization, and thick eccentric walls [15]. Acute DVT was evaluated using the four-point scale used for the overall CT image quality. Stent and prosthesis artifacts were scored using the following four-point scale: 1) strong streak artifacts with nondiagnostic insufficient image quality, 2) severe artifacts causing uncertainty, 3) mild artifacts with adequate image evaluation, and 4) excellent image quality with no visible artifacts.

## Objective image quality analysis

One blinded radiologist drew a circular region of interest (size, 1–3 cm$^2$) at the specific levels of the three axial images using PACS. The levels were IVC and midportions of right FV. The mean and standard deviation in Hounsfield units of the region of interest (i.e., attenuation, image noise) were calculated.

## Reference standards

Lesions from previous interventional venography for thrombectomy and/or ultrasound results and clinical date from electronic medical records were used.

## Radiation dose

The CTDI$_{vol}$ and dose–length product were described on the CT dose report to analyze the radiation dose [6, 12].

## Statistical analyses

The radiation dose and image analysis were compared between the three image sets using a one-way analysis of variance with post-hoc analysis and Bonferroni correction for multiple comparisons. A $p$-value below 0.05 was considered statistically significant. The statistical analyses were performed using SPSS version 21.0 (IBM, Armonk, NY, USA).

# Results

## Patients

The 100 patients who underwent the 105 examinations had a weight of 65.5±13.6 (range, 40.0–106.0 kg) and a body mass index of 24.8 ± 4.0 kg/m$^2$ (range, 12.0–32.0 kg/m$^2$) at the time of their corresponding examinations.

In the CT venography examinations, 24 patients presented DVT in 69 segments. Specifically, 10, 5, 4, 3, 1, and 1 patients showed DVT in 3, 2, 1, 5, 6, and 4 venous segments, respectively. In addition, 13 patients presented chronic venous change in 25 venous segments, while 10 patients presented varicose veins in 25 venous segments incidentally, and 17 patients presented the May–Thurner syndrome in 20 examinations. Moreover, 32 patients had a total of 48 metal prostheses affecting 77 venous segments with metal artifacts, corresponding to IVC filter ($n$ = 10), internal fixation of bone ($n$ = 8; femur, 6; tibia, 2), vertebroplasty ($n$ = 7), posterior lumbar interbody fusion ($n$ = 5), total hip replacement (THR; $n$ = 4; left, 3; right, 1), and total knee replacement (TKR; $n$ = 14; right, 7; left, 7). Seventeen patients had 18 stents (15 left CIV, 1 left FV, 2 right FV) appearing in 19 examinations (2 overlapping examinations).

## Subjective image quality analysis

The overall image quality of the standard-, low-, and ultralow-dose CT scans were scored at $4.0 \pm 0.1$ (range, 3–4), $4.0 \pm 0.2$ (range, 3–4), and $3.5 \pm 0.5$ (range, 3–4), respectively. The differences in segmental image quality between images were the largest in the IVC ($3.9 \pm 0.4$, $3.8 \pm 0.4$, $3.5 \pm 0.6$; $p < 0.001$), whereas no differences occurred between the three image sets for the popliteal and calf veins ($3.8 \pm 0.7$, right popliteal vein; $p = 0.977$). The scores of venous segments from the three image sets were 2–4 (adequate–excellent), except for a few popliteal veins. All calf veins showed scores of 3–4 in the three image sets, except for a right calf vein that scored 2 for ultralow-dose CT. The venous contrast quality showed scores of $4.0 \pm 0.1$ (range, 3–4), $4.0 \pm 0.1$ (range, 3–4), and $3.9 \pm 0.7$ (range, 3–4) for standard-, low-, and ultralow-dose CT, respectively. The detailed scores for the segments are described in **Table 2**.

The image quality of the 69 DVT segments showed higher scores for standard- and low-dose CT ($4.0 \pm 0.2$) than for ultralow-dose CT ($3.9 \pm 0.4$), as detailed in **Table 3**. All DVT segments for standard- and low-dose CT scored 3–4 (**Fig 2**) and only 2 segments (IVC) showed a score of 2 for ultralow-dose CT. Chronic venous change in 25 segments scored 4 for standard- and low-dose CT, and only 1 segment scored 3 for ultralow-dose CT. The varicose veins in 25 venous segments scored 4 on the three image sets.

The 48 metal prostheses produced artifacts in 77 venous segments, as detailed in **Table 4** (**Fig 3**). The abovementioned 13 segments of popliteal veins (6 right popliteal veins and 7 left popliteal veins) showed identically poor scores of 1, being unsuitable for diagnosis due to the artifacts from the metal prostheses for TKR. In addition, 29 segments scored 2 for ultralow-dose CT and 3 segments scored 2 for standard- and low-dose CT. The 18 stents in 19 examinations (2 overlapping examinations for the same patient) from 17 patients scored 4 in the three image sets, as detailed in **Table 5**.

**Table 2. Subjective and objective image quality of standard-, low-, and ultralow-dose CT venography scans.**

|  | Standard | Low | Ultralow |
|---|---|---|---|
| Subjective image quality |  |  |  |
| Overall image quality | 4.0 ± 0.1 | 4.0 ± 0.2 | 3.5 ± 0.5 |
| Segmental vein quality |  |  |  |
| Inferior vena cava | 3.9 ± 0.4 | 3.8 ± 0.4 | 3.5 ± 0.6 |
| Right common iliac vein | 3.9 ± 0.2 | 3.9 ± 0.3 | 3.8 ± 0.5 |
| Left common iliac vein | 3.9 ± 0.3 | 3.9 ± 0.3 | 3.8 ± 0.5 |
| Right femoval vein | 4.0 ± 0.3 | 4.0 ± 0.3 | 3.9 ± 0.4 |
| Left femoral vein | 3.9 ± 0.3 | 3.9 ± 0.3 | 3.8 ± 0.4 |
| Right popliteal vein | 3.8 ± 0.7 | 3.8 ± 0.7 | 3.8 ± 0.7 |
| Left popliteal vein | 3.8 ± 0.7 | 3.8 ± 0.7 | 3.8 ± 0.7 |
| Right calf vein | 3.9 ± 0.3 | 3.9 ± 0.3 | 3.9 ± 0.5 |
| Left calf vein | 3.9 ± 0.2 | 3.9 ± 0.2 | 3.9 ± 0.5 |
| Hounsfield unit |  |  |  |
| Attenuation |  |  |  |
| Inferior vena cava | 196.1 ± 31.2 | 195.0 ± 33.0 | 197.4 ± 31.6 |
| Left femoral vein | 183.4 ± 29.0 | 185.0 ± 28.9 | 181.0 ± 29.1 |
| Image noise |  |  |  |
| Inferior vena cava | 9.3 ± 2.3 | 11.2 ± 2.6 | 16.3 ± 3.7 |
| Left femoral vein | 7.4 ± 2.5 | 9.1 ± 3.1 | 11.1 ± 3.5 |

Note. Data are means ± standard deviations.

**Table 3. Subjective image quality of DVT segments on standard-, low-, and ultralow-dose CT venography scans.**

| Patient No. | Age /sex | Venous segment | Standard | Low | Ultralow |
|---|---|---|---|---|---|
| 1 | F/81 | Both calf vein | 4 | 4 | 4 |
| 2 | M/52 | LT PV, LT calf vein | 4 | 4 | 4 |
| 3 | M/71 | RT FV | 4 | 4 | 3 |
| | | RT PV, RT calf vein | 4 | 4 | 4 |
| 4 | M/37 | Both FV, Both PV | 4 | 4 | 4 |
| 5 | M/43 | RT PV, RT calf vein | 4 | 4 | 4 |
| 6 | F/48 | LT FV, LT PV, LT calf vein | 4 | 4 | 4 |
| 7 | M/72 | RT FV, RT PV, RT calf vein | 4 | 4 | 4 |
| 8 | F/60 | LT PV, LT calf vein | 4 | 4 | 4 |
| 9 | M/60 | RT FV, RT PV, RT calf vein | 4 | 4 | 4 |
| 10 | M/70 | LT FV, LT PV, LT calf vein | 4 | 4 | 4 |
| 11 | F/84 | LT EIV | 4 | 4 | 4 |
| | | LT FV | 3 | 3 | 3 |
| 12 | F/81 | Both FV, both PV, both calf vein | 4 | 4 | 4 |
| 13 | M/65 | RT FV | 4 | 4 | 3 |
| | | Both calf vein | 4 | 4 | 4 |
| 14 | M/68 | IVC | 3 | 3 | 2 |
| 15 | M/60 | RT FV, RT PV, RT calf vein | 4 | 4 | 4 |
| 16 | M/58 | LT FV, LT PV, LT calf vein | 4 | 4 | 4 |
| 17 | F/57 | LT calf vein | 4 | 4 | 4 |
| 18 | F/44 | RT FV, RT PV, RT calf vein | 4 | 4 | 4 |
| 19 | M/71 | RT FV | 4 | 4 | 3 |
| | | Both PV, both calf vein | 4 | 4 | 4 |
| 20 | M/43 | IVC | 3 | 3 | 2 |
| | | RT CIV, both EIV, RT FV | 4 | 4 | 4 |
| 21 | F/78 | LT CIV, LT EIV | 4 | 4 | 3 |
| | | LT FV, LT PV, LT calf vein | 4 | 4 | 4 |
| 22 | M/36 | LT CIV | 4 | 4 | 4 |
| 23 | M/68 | LT calf vein | 4 | 4 | 4 |
| 24 | M/70 | LT FV, LT PV, LT calf vein | 4 | 4 | 4 |

Abbreviations: F, female; M, male; LT, left; RT, right; IVC, inferior vena cava; CIV, common iliac vein; FV, femoral veins; PV, popliteal vein.

## Objective image quality analysis

The segments showed significantly higher image noise in the left femoral vein and IVC for ultralow-dose CT than for standard- and low-dose CT ($p < 0.001$). The noise levels in segments of the left femoral vein were 7.4 ± 2.5, 9.1 ± 3.1, and 11.1 ± 3.5 for standard-, low-, and ultralow-dose CT, respectively, while those of the IVC were 9.3 ± 2.3, 11.2 ± 2.6, and 16.3 ± 3.7, respectively. The differences in image noise between image sets were larger for the IVC than for the femoral vein. The objective image quality results are listed in **Table 2**.

## Radiation dose

The mean $CTDI_{vol}$ values for standard-, low-, and ultralow-dose CT were 4.4 ± 0.6, 3.1 ± 0.4, and 1.3 ± 0.2 mGy, respectively. The dose–length products for standard-, low-, and ultralow-dose CT were 567.9 ± 103.0, 397.5 ± 72.1, and 170.4 ± 30.9 mGy·cm, respectively. The mean $CTDI_{vol}$ and dose–length product for standard-dose CT showed significantly higher than those for low- and ultralow-dose CT ($p < 0.001$).

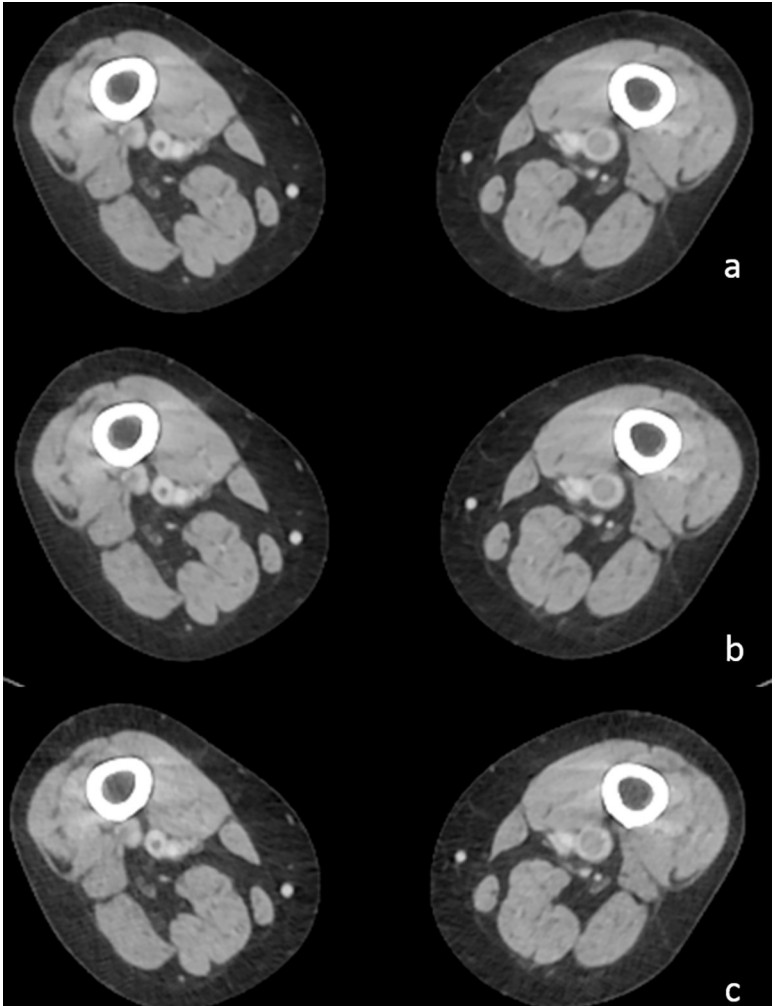

**Fig 2. CT venograms at 80 kVp of 81-year-old woman with DVT (body mass index, 26.1 kg/m²).** (a) Standard-dose (CTDI_vol, 4.6 mGy; dose–length product—DLP, 557.3 mGy·cm), (b) low-dose (CTDI_vol, 3.2 mGy; DLP, 390.1 mGy·cm), and (c) ultralow-dose (CTDI_vol, 1.4 mGy; DLP, 167.2 mGy·cm) CT scans show acute DVT in both popliteal veins. The scores of venous segments from the three image sets were 4.

## Discussion

Our study revealed similar subjective image quality for DVT, popliteal veins, calf veins, and metal artifacts on standard-, low-, and ultralow-dose CT venograms. Although standard-dose CT showed higher overall image quality than low- and ultralow-dose CT, reduced-dose CT venography (CTDI_vol, 1.3 mGy) provided a suitable image quality to evaluate DVT and lower-extremity veins when applying ADMIRE at 80 kVp. Previous studies have reported that CT venography at 80 kVp can reduce the radiation dose while maintaining image quality [6, 8, 16]. However, a detailed analysis regarding specific segmental veins, DVT, or metal artifacts has not been conducted. In this study, we investigated whether reduced-dose CT affects clinically important factors for DVT diagnosis and venogram evaluation.

The largest score differences in subjective image quality between standard-, low-, and ultralow-dose CT were found in the abdominal area corresponding to the segmental images of the IVC. Such differences originate from the theory that large solid organs (e.g., lower abdomen)

**Table 4. Subjective image quality of venous segments in 32 patients with metal prostheses on standard-, low-, and ultralow-dose CT venography.**

| Patient No. | Age /sex | Metal prosthesis | Affecting venous segment | Standard | Low | Ultralow |
|---|---|---|---|---|---|---|
| 1 | F/81 | S1 VP | RT CIV | 4 | 4 | 4 |
| | | | LT CIV | 4 | 3 | 3 |
| 2 | M/71 | IVC filter | IVC | 3 | 3 | 3 |
| 3 | F/48 | IVC filter | IVC | 3 | 3 | 3 |
| 4 | F/69 | T12 VP | IVC | 3 | 3 | 2 |
| 5 | M/59 | L4-5 PLIF | IVC, both CIV | 3 | 3 | 2 |
| 6 | F/62 | L3-5 PLIF | Both CIV | 3 | 3 | 2 |
| | | RT TKR | RT PV | 2 | 2 | 2 |
| 7 | M/79 | L3-5 PLIF | Both CIV | 4 | 3 | 2 |
| | | IVC filter | IVC | 3 | 3 | 3 |
| 8 | F/84 | LT femur IF | LT FV | 3 | 3 | 3 |
| 9 | M/68 | L3-5 PLIF | Both CIV | 4 | 3 | 3 |
| | | IVC filter | IVC | 3 | 3 | 2 |
| | | LT THR | LT EIV, LT FV | 3 | 3 | 3 |
| 10 | M/83 | L2 VP | IVC | 3 | 3 | 3 |
| 11 | F/88 | T12 VP | IVC | 4 | 4 | 4 |
| 12 | M/57 | RT tibia IF | RT calf vein | 2 | 2 | 2 |
| 13 | F/80 | IVC filter | IVC | 3 | 3 | 3 |
| 14 | F/88 | IVC filter | IVC | 3 | 3 | 3 |
| 15 | F/85 | RT TKR | RT PV | 1 | 1 | 1 |
| | | | RT calf vein | 3 | 3 | 2 |
| | | LT TKR | LT PV | 1 | 1 | 1 |
| | | | LT calf vein | 3 | 3 | 2 |
| 16 | M/43 | IVC filter | IVC | 3 | 3 | 3 |
| 17 | F/67 | L5-S1 PLIF | Both CIV | 3 | 3 | 3 |
| | | RT TKR | RT PV | 1 | 1 | 1 |
| | | | RT calf vein | 3 | 3 | 2 |
| | | LT TKR | LT PV | 1 | 1 | 1 |
| | | | LT calf vein | 3 | 3 | 2 |
| 18 | M/73 | IVC filter | IVC | 3 | 3 | 3 |
| 19 | F/84 | T12/L3 VP | IVC | 4 | 4 | 3 |
| | | RT TKR | RT PV | 1 | 1 | 1 |
| | | | RT calf vein | 3 | 3 | 2 |
| | | LT TKR | LT PV | 1 | 1 | 1 |
| | | | LT calf vein | 3 | 3 | 2 |
| 20 | F/30 | LT femur IF | LT FV | 4 | 3 | 3 |
| 21 | M/65 | LT THR | LT EIV, LT FV | 3 | 3 | 2 |
| 22 | F/84 | RT THR | RT EIV, RT FV | 3 | 3 | 2 |
| 23 | M/71 | IVC filter | IVC | 3 | 3 | 3 |
| | | RT femur IF | RT FV | 3 | 3 | 3 |
| 24 | F/76 | RT TKR | RT PV | 1 | 1 | 1 |
| | | | RT calf vein | 3 | 3 | 2 |
| | | LT TKR | LT PV | 1 | 1 | 1 |
| | | | LT calf vein | 3 | 3 | 2 |
| 25 | F/94 | IVC filter | IVC | 3 | 3 | 3 |
| | | RT TKR | RT PV | 1 | 1 | 1 |
| | | | RT calf vein | 3 | 3 | 2 |

*(Continued)*

**Table 4.** (Continued)

| Patient No. | Age /sex | Metal prosthesis | Affecting venous segment | Standard | Low | Ultralow |
|---|---|---|---|---|---|---|
| | | LT TKR | LT PV | 1 | 1 | 1 |
| | | | LT calf vein | 3 | 3 | 2 |
| 26 | M/28 | RT femur IF | RT FV | 2 | 2 | 2 |
| | | | RT PV | 3 | 3 | 3 |
| 27 | F/66 | RT TKR | RT PV | 1 | 1 | 1 |
| | | | RT calf vein | 3 | 3 | 3 |
| | | LT TKR | LT PV | 1 | 1 | 1 |
| | | | LT calf vein | 3 | 3 | 3 |
| 28 | M/36 | LT femur IF | LT FV | 3 | 3 | 2 |
| | | | LT PV | 3 | 3 | 3 |
| 29 | F/70 | L1-5 VP | IVC | 3 | 3 | 3 |
| | | | Both CIV | 3 | 3 | 2 |
| | | LT THR | LT EIV, LT FV | 3 | 3 | 3 |
| | | RT tibia IF | RT PV, RT calf vein | 3 | 3 | 3 |
| 30 | M/76 | LT femur IF | LT FV | 3 | 3 | 3 |
| 31 | F/80 | LT TKR | LT PV | 1 | 1 | 1 |
| | | | LT calf vein | 3 | 3 | 3 |
| 32 | F/86 | T12 VP | IVC | 3 | 3 | 3 |

Abbreviations: F, female; M, male; VP, vertebroplasty; PLIF, posterior lumbar interbody fusion; TKR, total knee replacement; THR, total hip replacement; IF, internal fixation; LT, left; RT, right; IVC, inferior vena cava; CIV, common iliac vein; FV, femoral veins; PV, popliteal vein.

require a high tube current using automatic tube current modulation according to the longitudinal (*z*-axis) mAs modulation [17]. On the other hand, the tube current can be reduced without a significant increase in the overall image noise in small body regions. Hence, CT scans of extremity veins show less beam attenuation than those of the abdomen, and CT scans of lower-extremity veins reflect suitable diagnostic image quality even when using low tube current for ultralow-dose CT. As a result, popliteal and calf veins showed no differences in segmental image quality between standard- and ultralow-dose CT. As the development of most DVT cases occurs in lower extremities with venous abnormality, our results support the use of reduced-dose CT venography applying ADMIRE at 80 kVp.

Lower CT tube voltages yield reduced radiation exposure but increased image noise [18]. Nevertheless, iterative reconstruction algorithms can minimize noise and provide a more acceptable image quality than filtered back-projection. Recent model-based iterative reconstruction algorithms enable direct reconstruction from raw data. However, previous studies have reported that model-based iterative reconstruction is time-consuming during its early stage [6, 19, 20], being unsuitable for the clinical workflow. In contrast, ADMIRE allows real-time CT scan reconstruction, contributing to the adoption of reconstruction in clinical settings. Moreover, advances in hardware equipped with Stellar detectors (Siemens Healthineers), which can reduce electronic noise by blending an analog digital converter chip to directly deliver a digital signal, can foster image quality while reducing the radiation dose for CT imaging at 80 kVp [21].

A concern about iterative reconstruction was related to the masking or underestimation of small lesions due to lesions with a low attenuation difference compared with surrounding tissue [22]. However, model-based iterative reconstruction provides more accuracy than statistical iterative reconstruction in the detection of small lesions in the abdomen while reducing

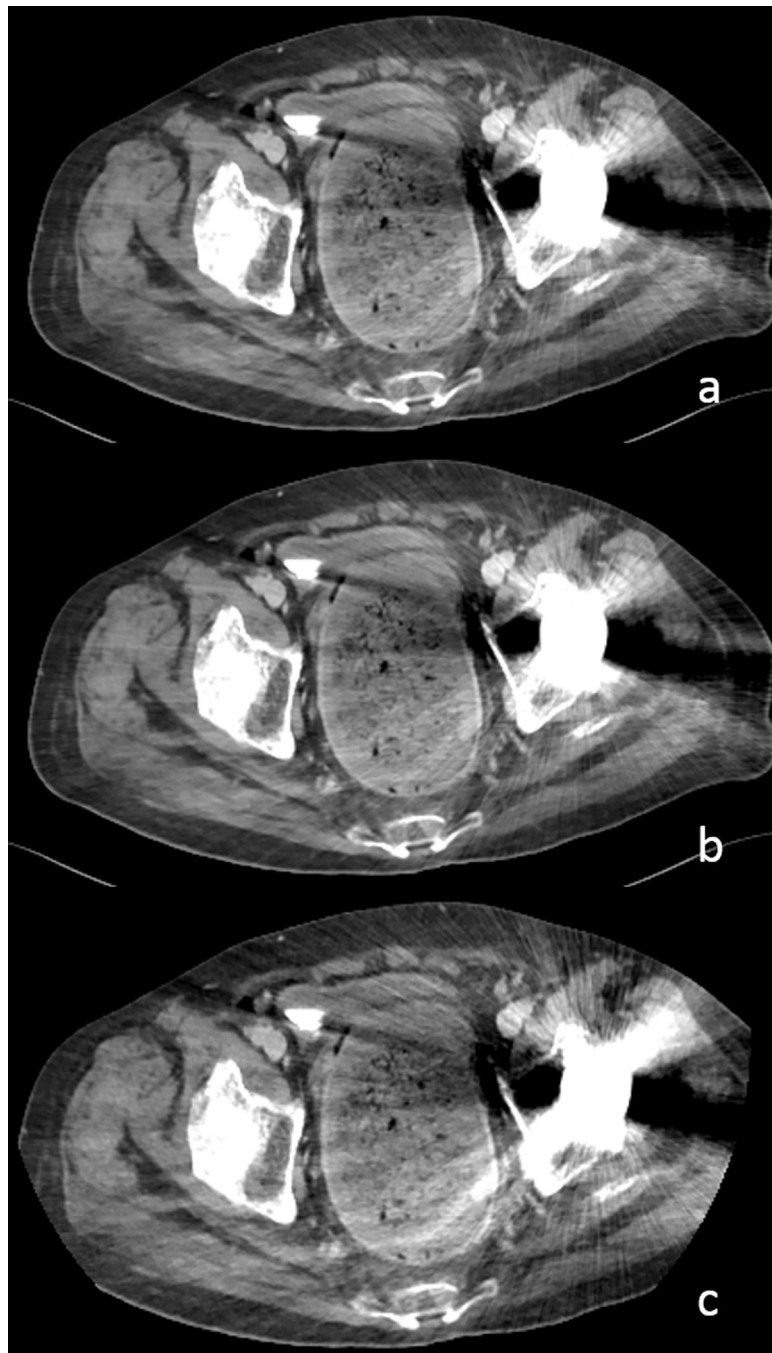

**Fig 3. CT venograms at 80 kVp of 65-year-old man (body mass index, 21.8 kg/m$^2$).** (a) Standard-dose (CTDI$_{vol}$, 6.0 mGy; DLP, 836.2 mGy·cm), (b) low-dose (CTDI$_{vol}$, 4.0 mGy; DLP, 585.3 mGy·cm), and (c) ultralow-dose (CTDI$_{vol}$, 2.0 mGy; DLP, 250.9 mGy·cm) CT scans. The segmental image quality of the left external iliac vein scored 3 for low- and standard-dose CT. Metal artifacts caused by THR affected the left iliac vein evaluation, reducing the score to 2 for ultralow-dose CT.

radiation dose and maintaining the image quality [23, 24]. Similarly, our results showed a comparable subjective image quality of small lesions (e.g., DVT, metal artifacts, stents) in lower extremities between standard- and low-dose CT.

**Table 5. Subjective image quality of venous segments in 17 patients with stent placement.**

| Patient No. | Age /sex | Venous segment | Stent patency | Standard | Low | Ultralow |
|---|---|---|---|---|---|---|
| 1 | F/48 | LT CIV | ISR | 4 | 4 | 4 |
| 2 | F/59 | LT CIV | Patent | 4 | 4 | 4 |
| 3–1), 2) | F/60 | LT CIV | ISR | 4 | 4 | 4 |
| 4 | F/51 | LT CIV | ISR | 4 | 4 | 4 |
| 5 | F/49 | LT CIV | ISR | 4 | 4 | 4 |
| 6 | M/79 | RT FV | ISR | 4 | 4 | 4 |
| 7 | M/68 | RT FV | ISR | 4 | 4 | 4 |
| 8 | F/88 | LT CIV | ISR | 4 | 4 | 4 |
| 9 | M/47 | LT CIV | Patent | 4 | 4 | 4 |
| 10–1), 2) | F/62 | LT CIV | ISR | 4 | 4 | 4 |
| 11 | F/80 | LT CIV | ISR | 4 | 4 | 4 |
| 12 | M/43 | LT CIV | ISR | 4 | 4 | 4 |
| 13 | F/68 | LT CIV | Patent | 4 | 4 | 4 |
| 14 | F/30 | LT CIV | ISR | 4 | 4 | 4 |
| 15 | F/78 | LT CIV | Occlusion | 4 | 4 | 4 |
| 16 | M/36 | LT CIV, LT FV | Occlusion | 4 | 4 | 4 |
| 17 | F/80 | LT CIV | Patent | 4 | 4 | 4 |

* 3–1), 2) and 10–1), 2) indicate that each patient underwent two examinations, respectively.

Abbreviations: F, female; M, male; LT, left; RT, right; CIV, common iliac vein; FV, femoral veins; ISR, in-stent restenosis.

Metal artifacts can degrade small lesion detection on CT scans [25]. Although most segmental veins showed acceptable/excellent image quality in the images with the 47 metal prostheses, 13 prostheses led to poor subjective image quality regardless of the radiation dose. These 13 prostheses correspond to TKR and affected the image quality in the popliteal veins. Thus, popliteal vein thrombosis may be underestimated in patients with metal prostheses in the knee joint regardless of the radiation dose. In these cases, ultrasound may be more suitable for accurate DVT diagnosis than CT venography.

This study has some limitations. First, this was a retrospective study considering CT venography examinations, which has a selection bias. Second, the examinations were conducted on relatively only a patient with severe obesity, which undermines imaging quality. The results of our study do not directly translate to the severely obese patients. Third, we did not analyze interobserver variability for subjective image analysis or diagnostic performance for DVT detection. Fourth, we selected fixed 80 kVp and compared between specific radiation doses. Selection of automatic kVp change or fixed kVp is possible in Siemens CT. However, using the specific split of the tube dose in a dual source mode, we can only select a specific kVp (i.e, cannot use automatic kVp change). Finally, image quality compared to that using other tube voltages (e.g., 70, 90, and 100 kVp) or other image reconstruction methods (i.e., filtered back projection) was not assessed. These limitations hinder the generalization of our results toward the widespread use of low-dose CT venography.

Overall, our results suggest the low- and ultralow-dose CT venography at 80 kVp using ADMIRE show acceptable image quality for DVT evaluation and follow-up.

## Supporting information

**S1 Appendix. STROBE checklist.**
(DOC)

## Acknowledgments

The authors thank Seong Yong Pak, for his help in reconstructing the image data.

## Author Contributions

**Conceptualization:** Jung Han Hwang.

**Data curation:** Jin Mo Kang, Sang tae Choi.

**Formal analysis:** Jung Han Hwang, Suyoung Park.

**Funding acquisition:** So Hyun Park.

**Methodology:** So Hyun Park.

**Project administration:** Jeong Ho Kim.

**Resources:** Jin Mo Kang.

**Validation:** Jeong Ho Kim.

**Writing – original draft:** So Hyun Park.

**Writing – review & editing:** Suyoung Park.

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
