## [Decision Letter · Decision Letter 0]

26 May 2021

PONE-D-21-10760

Comparison study of image quality at various radiation doses for CT venography using advanced modeled iterative reconstruction

PLOS ONE

Dear Dr. Park,

Thank you for submitting your manuscript to PLOS ONE. After careful consideration, we feel that it has merit but does not fully meet PLOS ONE’s publication criteria as it currently stands. Therefore, we invite you to submit a revised version of the manuscript that addresses the points raised during the review process. More details of the points raised by the reviewers and editor can be found in their comments at the end of this letter.

We look forward to receiving your revised manuscript.

Kind regards,

Mingwu Jin, Ph.D.

Academic Editor

PLOS ONE

Additional Editor Comments:

The reviewers raised the major concerns about the feasibility of CT protocol and the lack of comparison of bench-mark methods, such as FBP. The latter may be necessary to establish that the model-based iterative reconstruction algorithm (ADMIRE) is essential for low-dose CT venogram. In addition, the readers' experience and performance difference should be provided for the subjective evaluation. Addressing these issues would greatly improve the significance of the paper.

Journal Requirements:

Reviewers' comments:

Reviewer's Responses to Questions

**Comments to the Author**

1. Is the manuscript technically sound, and do the data support the conclusions?

Reviewer #1: No

Reviewer #2: No

2. Has the statistical analysis been performed appropriately and rigorously? 

Reviewer #1: No

Reviewer #2: Yes

3. Have the authors made all data underlying the findings in their manuscript fully available?

Reviewer #1: Yes

Reviewer #2: Yes

4. Is the manuscript presented in an intelligible fashion and written in standard English?

Reviewer #1: Yes

Reviewer #2: Yes

5. Review Comments to the Author

Reviewer #1: A review of the paper titled “Comparison study of image quality at various radiation doses for CT venography using advanced modeled iterative reconstruction”

Thank for letting me review this work! It is good to see we can go lower in dose using this scanner/recon for these indications and not suffer issues.

General Comments:

1. The abstract needs some attention. The conclusion statement says “using a reconstruction algorithm allows” I think you need to say “using an advanced model based iterative reconstruction algorithm”

2. It is not clear from the abstract that you compare regular and lower dose non advanced recon to lower dose advanced recon. You need to do this to support your conclusions to prove it was the advanced recon that enabled the dose reduction. Or else how to do we know the overall dose was just set too high? I assume you did this in the paper itself, but reading the abstract itself, this important study design point is not clear. For example, I can scan at 100 mGy using FBP and then cut it to 50 mGy and use ADMIRE. But concluding ADMIRE allowed a reduction w/o issues for diagnostic purposes is flawed…I would need to show at 50 mGy with FBP I performed worse.

1. Introduction: I don’t think this sentence means what you intend “The radiation exposure during CT venography is becoming an increasing concern 67 because the mean dose level rises when a patient requires multiple CT examinations during the 68 disease process of pulmonary embolism (chest CT) or given the recurrency of DVT” There is no reason for the mean dose the pt gets to change, I think you mean their cumulative effective dose is rising as they get more and more scans? Using CED as a motivation for dose reduction is kind of controversial, an exam, if indicated and appropriate should be given w/o concern for stochastic cancer risk. This is the position of the American College of Radiology and the American Association of Physicists in Medicine.

2. Introduction: I think there a lot of papers out there for this, please cite more (I went to google scholar and found several on the first page of results you don’t cite on this topic) and consider changing the verbiage/tone of this paragraph, to something more like “there are many papers in the literature looking at dose reduction with lower kV and advanced model based recons…we are contributing to this space by looking specifically at …

a. “Few 69 studies have reported the reduction of the average radiation dose for CT venography using low 70 tube voltages, mono-energy CT, and iterative reconstruction algorithms”

3. There is no stat testing for the reader scores/ Just for the noise measurements?

4. The paper concludes it is recon method that enables the low dose visualization, but we cannot conclude this from your methods. The scores seem to basically be the same or just differ by 1 point from low to high dose, with more variation being pt to pt. I think in order to support that the lower dose was enabled by advanced recon, you must prove to us the images are not fit for diagnostic purposes if the ADMIRE recon was not used?

Specific Comments:

1. In the CT protocol section, can you elaborate on what the 70 and 30% refer to? Was the actual mAs equal to what you have listed, or was 70 and 30% used?

Reviewer #2: In this manuscript, the author evaluated image quality of three radiation dose levels for CT venography. The results showed low- and ultralow-dose CT venography at 80 kV had acceptable image quality.

Comments

1) My biggest question is that how it was possible for the author to run the Siemens Force scanner with both tubes at 80 kV and different tube currents. For Siemens Force scanner, all the clinical protocols using two sources are in either dual-energy mode or dual-source mode. The dual-energy mode has one tube high kV and one tube low kV. In dual-source mode, both tubes are using the same tube current. Did the author use some special service mode or collaborate with Siemens to config the protocols? Otherwise, it is not possible to scan using the protocols described in the manuscript.

2) Line 42. CT dose index volume is not “calculated” but displayed on the dose page. I don’t think there was any calculation about CTDIvol in the manuscript.

3) The abstract needs to mention the special setup of the protocols using dual-source CT.

4) Have the author tried the CarekV, an automatic kV selection tool used by Siemens CT? How does it compare to 80 kV method used in the manuscript?

6. PLOS authors have the option to publish the peer review history of their article (what does this mean?). If published, this will include your full peer review and any attached files.

Reviewer #1: No

Reviewer #2: No

---

## [Author Response · Author response to Decision Letter 0]

2 Jul 2021

Reviewer's Responses to Questions

Comments to the Author

1. Is the manuscript technically sound, and do the data support the conclusions?

Reviewer #1: No

Reviewer #2: No

2. Has the statistical analysis been performed appropriately and rigorously?

Reviewer #1: No

Reviewer #2: Yes

3. Have the authors made all data underlying the findings in their manuscript fully available?

Reviewer #1: Yes

Reviewer #2: Yes

4. Is the manuscript presented in an intelligible fashion and written in standard English?

Reviewer #1: Yes

Reviewer #2: Yes

5. Review Comments to the Author

Reviewer #1: A review of the paper titled “Comparison study of image quality at various radiation doses for CT venography using advanced modeled iterative reconstruction”

Thank for letting me review this work! It is good to see we can go lower in dose using this scanner/recon for these indications and not suffer issues.

General Comments:

1. The abstract needs some attention. The conclusion statement says “using a reconstruction algorithm allows” I think you need to say “using an advanced model based iterative reconstruction algorithm”

Response: Thank you for your comment. We have changed the abstract as suggested “Low- and ultralow-dose CT venography at 80 kVp using an advanced model based iterative reconstruction algorithm” (R1-1).

2. It is not clear from the abstract that you compare regular and lower dose non advanced recon to lower dose advanced recon. You need to do this to support your conclusions to prove it was the advanced recon that enabled the dose reduction. Or else how to do we know the overall dose was just set too high? I assume you did this in the paper itself, but reading the abstract itself, this important study design point is not clear. For example, I can scan at 100 mGy using FBP and then cut it to 50 mGy and use ADMIRE. But concluding ADMIRE allowed a reduction w/o issues for diagnostic purposes is flawed…I would need to show at 50 mGy with FBP I performed worse.

Response: Thank you for your comment. We were routinely inspecting the lower extremity CT venography using ADMIRE on a CT machine (Force). At that time, we conducted using caredose at 300 reference mAs when carekVp was set at 80kVp. It would have been a superior study comparing images with FBP reconstructing and those with ADMIRE regarding dose reduction. However, raw data work will be needed to reconstruct FBP images in comparison with ADMIRE images. We did not perform the work. We have added the limitation of the study in the manuscript (R1-2). Currently, major CT vendors introduced iterative reconstruction algorithms for clinical routine, which evolved rapidly into increasingly advanced reconstruction algorithms. Most dose-reduction strategies remained in the domain of decreasing tube current or tube voltage while iterative reconstruction algorithms insure an acceptable diagnostic image quality. Therefore, we focused on dose reduction study of intraindividual comparison using ADMIRE. 

1. Introduction: I don’t think this sentence means what you intend “The radiation exposure during CT venography is becoming an increasing concern 67 because the mean dose level rises when a patient requires multiple CT examinations during the 68 disease process of pulmonary embolism (chest CT) or given the recurrency of DVT” There is no reason for the mean dose the pt gets to change, I think you mean their cumulative effective dose is rising as they get more and more scans? Using CED as a motivation for dose reduction is kind of controversial, an exam, if indicated and appropriate should be given w/o concern for stochastic cancer risk. This is the position of the American College of Radiology and the American Association of Physicists in Medicine.

Response: Thank you for your comment. We have removed the sentence as suggested (R1-3)

2. Introduction: I think there a lot of papers out there for this, please cite more (I went to google scholar and found several on the first page of results you don’t cite on this topic) and consider changing the verbiage/tone of this paragraph, to something more like “there are many papers in the literature looking at dose reduction with lower kV and advanced model based recons…we are contributing to this space by looking specifically at …

a. “Few 69 studies have reported the reduction of the average radiation dose for CT venography using low 70 tube voltages, mono-energy CT, and iterative reconstruction algorithms”

Response) Thank you for your comment. We have changed the introduction and added several references as suggested (R1-4, There are many studies in the literature investigating at radiation dose reduction with low tube voltages and advanced model based reconstruction. We are contributing to this space by specifically looking at..). 

3. There is no stat testing for the reader scores/ Just for the noise measurements?

Response: Thank you for your comment. Two radiologists reviewed subjective image analysis (i.e. scores) in consensus. We did not analyze interobserver variability. We only measured attenuation and image noise. Therefore, we added the limitation of this study in the manuscript (R1-5). We analyzed differences of subjective image quality and image noise of three image sets using a one-way analysis of variance with post-hoc analysis and Bonferroni correction. 

4. The paper concludes it is recon method that enables the low dose visualization, but we cannot conclude this from your methods. The scores seem to basically be the same or just differ by 1 point from low to high dose, with more variation being pt to pt. I think in order to support that the lower dose was enabled by advanced recon, you must prove to us the images are not fit for diagnostic purposes if the ADMIRE recon was not used?

Response: Thank you for your comment. We did not analyze diagnostic performance for DVT detection or compare between images with filtered back projection and those with ADMIRE. We have added our study’s limitation (we did not analyze diagnostic performance for DVT detection) and change in the conclusion (~ADMIRE show acceptable image quality for DVT evaluation~, R1-6).

Specific Comments:

1. In the CT protocol section, can you elaborate on what the 70 and 30% refer to? Was the actual mAs equal to what you have listed, or was 70 and 30% used?

Response: We used 70% and 30% reference mAs, compared with reference mAs on standard CT. The actual mAs also showed 70% and 30% ratio (low-dose and ultralow-dose), compared with standard dose CT.

Reviewer #2: In this manuscript, the author evaluated image quality of three radiation dose levels for CT venography. The results showed low- and ultralow-dose CT venography at 80 kV had acceptable image quality.

Comments

1) My biggest question is that how it was possible for the author to run the Siemens Force scanner with both tubes at 80 kV and different tube currents. For Siemens Force scanner, all the clinical protocols using two sources are in either dual-energy mode or dual-source mode. The dual-energy mode has one tube high kV and one tube low kV. In dual-source mode, both tubes are using the same tube current. Did the author use some special service mode or collaborate with Siemens to config the protocols? Otherwise, it is not possible to scan using the protocols described in the manuscript.

Response: Thank you for your comment. To perform the specific split of the tube dose (mAs) between tube detector A and B, the CT scanner requires a dual energy research license. We added the feature in the material and method (R2-1).

2) Line 42. CT dose index volume is not “calculated” but displayed on the dose page. I don’t think there was any calculation about CTDIvol in the manuscript.

Response: Thank you for your comment. We changed the word, as “displayed” (R2-2).

3) The abstract needs to mention the special setup of the protocols using dual-source CT.

Response: Thank you for your comment. We added the protocol in methods section, “in the dual-source mode in tubes A (reference mAs, 210 mAs at 70%) and B (reference mAs, 90 mAs at 30%) at a fixed 80 kVp” (R2-3).

4) Have the author tried the CarekV, an automatic kV selection tool used by Siemens CT? How does it compare to 80 kV method used in the manuscript?

Response: Thank you for your comment. In Siemens CT, selection of automatic kVp or fixed kVp is possible. However, using the specific split of the tube dose in a dual source mode, we select a specific kVp (i.e, did not use automatic kVp change). Therefore, we selected fixed 80 kVp and compared between specific radiation doses.

---

## [Decision Letter · Decision Letter 1]

2 Aug 2021

PONE-D-21-10760R1

Comparison study of image quality at various radiation doses for CT venography using advanced modeled iterative reconstruction

PLOS ONE

Dear Dr. Park,

Thank you for submitting your manuscript to PLOS ONE. After careful consideration, we feel that it has merit but does not fully meet PLOS ONE’s publication criteria as it currently stands. Therefore, we invite you to submit a revised version of the manuscript that addresses the points raised during the review process.

Please re-address the comment 4 of Reviewer 2 about CarekV from last review. Since CarekV is a standard package on the scanner for the same purpose of this study, please add some discussion about the limitation of this study if the experiment suggested by the reviewer can't be conducted.

We look forward to receiving your revised manuscript.

Kind regards,

Mingwu Jin, Ph.D.

Academic Editor

PLOS ONE

Journal Requirements:

Reviewers' comments:

Reviewer's Responses to Questions

**Comments to the Author**

1. If the authors have adequately addressed your comments raised in a previous round of review and you feel that this manuscript is now acceptable for publication, you may indicate that here to bypass the “Comments to the Author” section, enter your conflict of interest statement in the “Confidential to Editor” section, and submit your "Accept" recommendation.

Reviewer #1: All comments have been addressed

Reviewer #2: (No Response)

2. Is the manuscript technically sound, and do the data support the conclusions?

Reviewer #1: Yes

Reviewer #2: Yes

3. Has the statistical analysis been performed appropriately and rigorously? 

Reviewer #1: Yes

Reviewer #2: Yes

4. Have the authors made all data underlying the findings in their manuscript fully available?

Reviewer #1: Yes

Reviewer #2: Yes

5. Is the manuscript presented in an intelligible fashion and written in standard English?

Reviewer #1: Yes

Reviewer #2: Yes

6. Review Comments to the Author

Reviewer #1: The authors have addressed my concerns. The major issue I had was that we cannot claim it was the advanced recon that allowed the dose reduction, I think the authors have changed the paper's verbiage to account for the fact their study never compared FBP low dose to ADMIRE low dose directly. Thanks.

Reviewer #2: (No Response)

7. PLOS authors have the option to publish the peer review history of their article (what does this mean?). If published, this will include your full peer review and any attached files.

Reviewer #1: **Yes: **Timothy P Szczykutowicz

Reviewer #2: No

---

## [Author Response · Author response to Decision Letter 1]

5 Aug 2021

Editor

Please re-address the comment 4 of Reviewer 2 about CarekV from last review. Since CarekV is a standard package on the scanner for the same purpose of this study, please add some discussion about the limitation of this study if the experiment suggested by the reviewer can't be conducted.

Response: Thank you for your comment. We have added the limitation as suggested “Fourth, we selected fixed 80 kVp and compared between specific radiation doses. Selection of automatic kVp change or fixed kVp is possible in Siemens CT. However, using the specific split of the tube dose in a dual source mode, we can only select a specific kVp (i.e, cannot use automatic kVp change).” (Editor).

Journal Requirements:

Response: Thank you for your comment.

Reviewers' comments:

Reviewer's Responses to Questions

Comments to the Author

1. If the authors have adequately addressed your comments raised in a previous round of review and you feel that this manuscript is now acceptable for publication, you may indicate that here to bypass the “Comments to the Author” section, enter your conflict of interest statement in the “Confidential to Editor” section, and submit your "Accept" recommendation.

Reviewer #1: All comments have been addressed

Reviewer #2: (No Response)

2. Is the manuscript technically sound, and do the data support the conclusions?

Reviewer #1: Yes

Reviewer #2: Yes

3. Has the statistical analysis been performed appropriately and rigorously?

Reviewer #1: Yes

Reviewer #2: Yes

4. Have the authors made all data underlying the findings in their manuscript fully available?

Reviewer #1: Yes

Reviewer #2: Yes

5. Is the manuscript presented in an intelligible fashion and written in standard English?

Reviewer #1: Yes

Reviewer #2: Yes

6. Review Comments to the Author

Reviewer #1: The authors have addressed my concerns. The major issue I had was that we cannot claim it was the advanced recon that allowed the dose reduction, I think the authors have changed the paper's verbiage to account for the fact their study never compared FBP low dose to ADMIRE low dose directly. Thanks.

Reviewer #2: (No Response)

7. PLOS authors have the option to publish the peer review history of their article (what does this mean?). If published, this will include your full peer review and any attached files.

Do you want your identity to be public for this peer review? For information about this choice, including consent withdrawal, please see our Privacy Policy.

Reviewer #1: Yes: Timothy P Szczykutowicz

Reviewer #2: No

Response 1~7): Thank you for your helpful comment.

---

## [Editor Report · Decision Letter 2]

10 Aug 2021

Comparison study of image quality at various radiation doses for CT venography using advanced modeled iterative reconstruction

PONE-D-21-10760R2

Dear Dr. Park,

We’re pleased to inform you that your manuscript has been judged scientifically suitable for publication and will be formally accepted for publication once it meets all outstanding technical requirements.

Kind regards,

Mingwu Jin, Ph.D.

Academic Editor

PLOS ONE

---

## [Editor Report · Acceptance letter]

23 Aug 2021

PONE-D-21-10760R2 

Comparison study of image quality at various radiation doses for CT venography using advanced modeled iterative reconstruction 

Dear Dr. Park:

I'm pleased to inform you that your manuscript has been deemed suitable for publication in PLOS ONE. Congratulations! Your manuscript is now with our production department. 

Kind regards, 

on behalf of

Dr. Mingwu Jin 

Academic Editor

PLOS ONE